

# BLAST-based validation of metagenomic sequence assignments

Adam L. Bazinet[1], Brian D. Ondov[1,2], Daniel D. Sommer[1] and Shashikala Ratnayake[1]

[1] National Biodefense Analysis and Countermeasures Center, Fort Detrick, MD, USA
[2] National Human Genome Research Institute, Bethesda, MD, USA

## ABSTRACT

When performing bioforensic casework, it is important to be able to reliably detect the presence of a particular organism in a metagenomic sample, even if the organism is only present in a trace amount. For this task, it is common to use a sequence classification program that determines the taxonomic affiliation of individual sequence reads by comparing them to reference database sequences. As metagenomic data sets often consist of millions or billions of reads that need to be compared to reference databases containing millions of sequences, such sequence classification programs typically use search heuristics and databases with reduced sequence diversity to speed up the analysis, which can lead to incorrect assignments. Thus, in a bioforensic setting where correct assignments are paramount, assignments of interest made by "first-pass" classifiers should be confirmed using the most precise methods and comprehensive databases available. In this study we present a BLAST-based method for validating the assignments made by less precise sequence classification programs, with optimal parameters for filtering of BLAST results determined via simulation of sequence reads from genomes of interest, and we apply the method to the detection of four pathogenic organisms. The software implementing the method is open source and freely available.

## INTRODUCTION

In metagenomic analysis, comparing the genomic sequence content of a sample to a reference database is fundamental to understanding which organisms present in the sample were sequenced. There exist many bioinformatics software programs that perform this classification task (*Bazinet & Cummings, 2012*; *Breitwieser, Lu & Salzberg, 2017*; *McIntyre et al., 2017*; *Sczyrba et al., 2017*); some programs only estimate overall taxonomic composition and abundance in the sample (*Koslicki & Falush, 2016*; *Schaeffer et al., 2015*), while other programs assign a taxonomic label to each metagenomic sequence (*Huson et al., 2007*; *Ames et al., 2013*; *Wood & Salzberg, 2014*; *Hong et al., 2014*; *Ounit et al., 2015*; *Gregor et al., 2016*; *Kim et al., 2016*). In a bioforensic setting, one is often concerned with reliably detecting the presence of a particular organism in a metagenomic sample, which may only be present in a trace amount. For this task, one typically uses the

Corresponding author
Adam L. Bazinet,
adam.bazinet@nbacc.dhs.gov

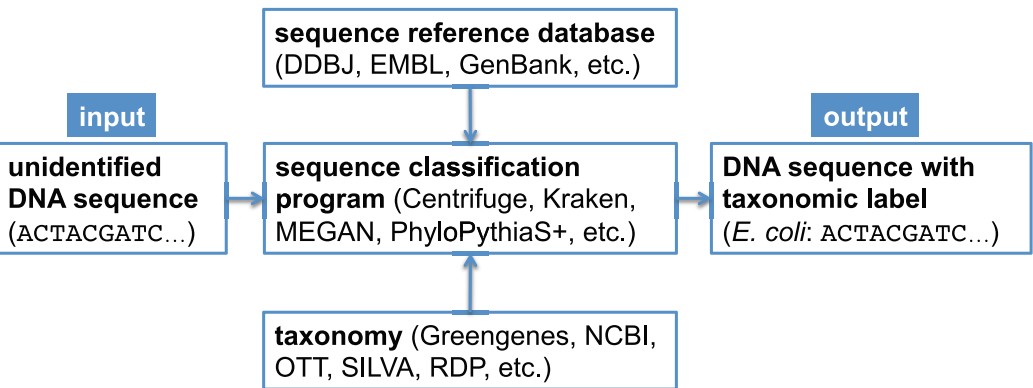

**Figure 1 Canonical workflow for the classification of metagenomic sequences.** A sequence classification program, which typically makes use of a reference database and a taxonomy, is used to assign taxonomic labels to unidentified DNA sequences.

latter class of programs just mentioned, which determine the taxonomic affiliation of each sequence using a reference database (*Mashima et al., 2017*; *Kulikova et al., 2004*; *Benson et al., 2014*) and a taxonomy (*Balvočiūtė & Huson, 2017*). A canonical metagenomic sequence classification workflow is shown in Fig. 1.

When classifying sequences, there is a general trade-off between sensitivity (the proportion of the total number of sequences assigned correctly) and precision (the proportion of assigned sequences assigned correctly), as well as between classification performance (combined sensitivity and precision) and computational resource requirements. Modern metagenomic sequence classification programs often use relatively fast heuristics and databases with limited sequence diversity to increase analysis speed, as metagenomic data sets often consist of millions or billions of sequences that need to be compared to millions of database sequences. Thus, while they are useful in performing a "first-pass" analysis, in a bioforensic setting it is important to validate the assignments of interest made by such programs using the most precise methods available (*Gonzalez et al., 2016*). One could choose to validate only the assignments made to the taxonomic clade of interest (e.g., *Bacillus anthracis*), but depending on the computing capacity one has access to, one might choose to validate all assignments subsumed by a higher ranking taxon (e.g., the *B. cereus* group or the *Bacillus* genus), which would enable the detection of possible false negative assignments as well as false positive assignments made by the first-pass classifier.

In this study, we present a method that uses BLAST (*Camacho et al., 2009*), the NCBI non-redundant nucleotide database (*NCBI Resource Coordinators, 2016*) (`nt`), and the NCBI taxonomy (*NCBI Resource Coordinators, 2016*) to validate the assignments made by less precise sequence classification programs. BLAST is widely considered the "gold standard" for sequence comparison, although it is generally known to be orders of magnitude slower than the most commonly used first-pass classifiers (see *Bazinet & Cummings, 2012* for a comparison of BLAST runtimes to those of other sequence classification programs). For simplicity, we refer to the taxonomic clade of interest in our

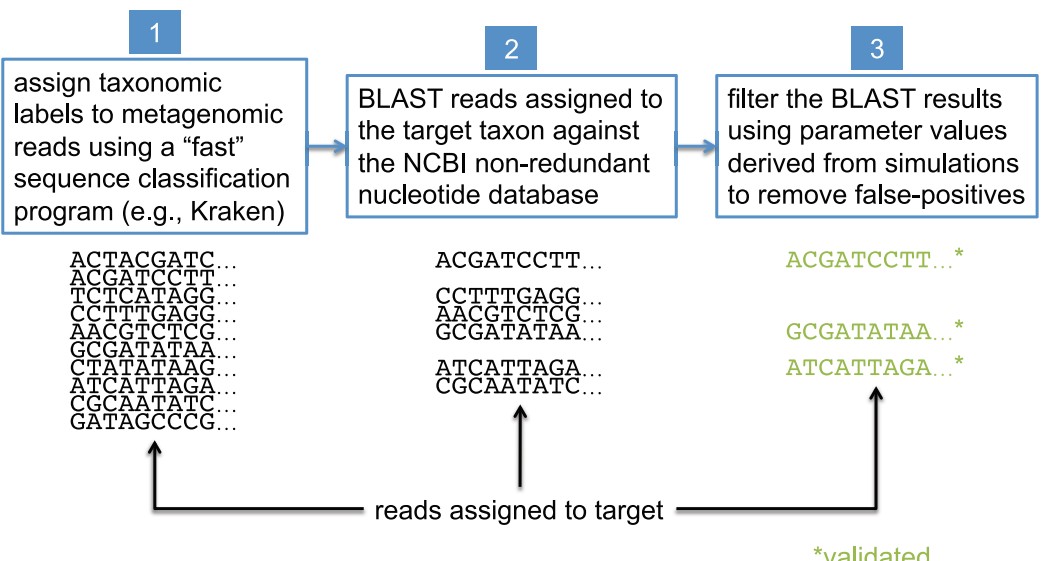

**Figure 2** **Workflow for BLAST-based validation of taxonomic assignments.** Taxonomic labels are first assigned to metagenomic reads using a "first-pass" classification program. Reads assigned to a target taxon of interest are then compared against the NCBI nt database using BLAST. Final taxonomic assignments are obtained by filtering the BLAST results using parameter values that were previously determined to be optimal for the target taxon.

analyses as the "target taxon," and we assume all metagenomic sequences are paired-end reads generated by the Illumina HiSeq 2500 sequencer (no assembled sequences). The BLAST-based validation procedure involves comparing each read against the `nt` database using BLASTN, and then filtering and interpreting the BLAST results based on data collected from simulated read experiments aimed at optimizing detection of the target taxon; this workflow is shown in Fig. 2. Before presenting additional details about the BLAST-based validation procedure, however, we first describe some related work from the literature.

## Related work

### Platypus Conquistador

An existing software tool, "Platypus Conquistador" (*Gonzalez et al., 2016*), also uses BLAST to validate the classification of particular sequences of interest. Platypus requires the user to split their reference sequences into two databases: a database containing only sequences of possible interest, and a database composed of potential "background" sequences. BLAST queries are performed against both databases, and hits may be filtered by various combinations of percent identity or alignment length values, which need to be provided by the user. After filtering, query sequences with hits to the "interest" database are checked to see if they also have hits to the "background" database; if so, the bit scores of the respective best hits are compared and are roughly categorized as "equal," "interest > background," etc. While this could be a helpful diagnostic tool, there is no guidance provided to the user as to what parameter values to use or what difference in bit scores between interest and background should be regarded as significant. Furthermore, this tool no longer appears to be actively developed.

### Genomic purity assessment

Whereas in this study we are concerned with the precision with which individual reads are classified so as to be confident in the detection of a target taxon in a metagenomic sample, a recently published study (*Olson et al., 2017*) addresses a different, but related problem, namely detecting contaminant organisms in ostensibly axenic (non-metagenomic) samples. Specifically, *Olson et al. (2017)* develop methods to determine the proportion of a contaminant required to be present in an otherwise pure material such that the contaminant can be detected with standard metagenomic sequence classification tools. As in our study, they simulate reads with ART (*Huang et al., 2011*) software (in their case from both "material" and potential contaminant genomes) to set up conditions under which sequence classification performance can be assessed. PathoScope (*Hong et al., 2014*) is used instead of BLAST for read classification. In general, they find that their method is able to identify contaminants present in a proportion of at least $10^{-3}$ for most contaminant-material combinations tested.

### Outlier detection in BLAST hits

*Shah, Altschul & Pop (2017)* have developed a method that detects outliers among BLAST hits in order to separate the hits most closely related to the query from hits that are phylogenetically more distant using a modified form of Bayesian Integral Log odds scores (*Altschul et al., 2010*) and a multiple alignment scoring function. In this way, they separate sequences with confident taxonomic assignments from sequences that should be analyzed further. The method was developed for and tested on 16S rRNA data, and thus is currently not applicable to whole genome sequencing data sets. As a general-purpose filter, however, it can be used with any organism containing 16S rRNA data, whereas our methods are optimized for detection of specific taxa. It is also interesting to note that in the Shah et al. study, BLAST is used as a first-pass classifier and subsequent analysis is performed with Taxonomic Identification and Phylogenetic Profiling (TIPP) (*Nguyen et al., 2014*), whereas in the paradigm we present here, a much faster classifier than BLAST would be used for a first-pass (e.g., Kraken (*Wood & Salzberg, 2014*)), and then our BLAST-based method would be used for validation.

## METHODS

### Use of a "first-pass" taxonomic classifier

We selected Kraken (*Wood & Salzberg, 2014*) (version 1.1) as the "first-pass" taxonomic classification program to be used in this study, primarily because of its widespread use in the bioinformatics community at large (the bioforensics community being no exception). Kraken was run in paired-end mode with default parameters and used standard Kraken databases for bacteria, archaea, viruses, plasmids, and human sequences.

### Read simulation

For read simulation we used ART (*Huang et al., 2011*) (version 2016-06-05). To ensure thorough sampling, all experiments used simulated reads equivalent in total to $10\times$ coverage of the source genome. For 150-bp reads, we used the built-in HS25 quality

profile with an insert size of 200 ± 10 bp (mean ± standard deviation). For 250-bp reads, we used a custom Illumina HiSeq quality profile that we generated from recent runs of our HiSeq 2500, with an insert size of 868 ± 408 bp determined from recent library preparations. We supplied this information so that the simulated reads would have characteristics that closely matched what we would expect to obtain from a real HiSeq run in our laboratory, thus ensuring that the simulation results would be maximally useful to us. We recommend that others who emulate our procedures customize the attributes of their simulated reads to correspond to the real data they anticipate analyzing.

## Similarity searches

We used BLASTN from the BLAST+ package (*Camacho et al., 2009*) (version 2.2.25+) together with the NCBI non-redundant nucleotide database (*NCBI Resource Coordinators, 2016*) (downloaded February 2017) for all classification experiments. Default parameters were used, except when excluding taxa from the reference database, in which case the -negative_gilist option was added. The BLAST computation was distributed over many cluster nodes to complete the analyses in a timely manner.

## BLAST result filters

The output from BLAST includes a number of statistics that can potentially be used to filter the results, including alignment length, alignment percent identity, *E*-value (the number of similar scoring alignments one can "expect" to see by chance in a database of the size being searched), and bit score (a database size-independent measure of alignment quality).

We developed two basic ways of filtering BLAST results. The first we term an "absolute" filter, which simply removes BLAST hits that do not meet a particular criterion. Various possible criteria include minimum alignment length, minimum alignment percent identity, or maximum *E*-value. Of these three filters, this study only uses the *E*-value filter (abbreviated *E*), as *E*-value is fundamentally a composite of alignment length and alignment similarity. (Our software supports the use of all three filters, however, either individually or in combination.) If the best BLAST hit matches the target taxon after application of the absolute filters, it is then possible to apply a "relative" filter by computing the difference in *E*-value or bit score between the best hit and the best hit to a *non-target* taxon (should the latter exist). As very small *E*-values are typically rounded to zero, our software uses relative bit scores in this context for maximum applicability; we call this quantity the "bit score difference," abbreviated *b*. If *b* is greater than or equal to a threshold determined via read simulation experiments, then we have validated the assignment of the read to the target taxon. Examples of the application of the bit score difference filter are given in Figs. 3 and 4.

## Evaluation of classification performance

The two main metrics used in this study to evaluate classification performance are sensitivity and precision. The formula we use for sensitivity is slightly different from the standard one, though, and we also use a non-standard formula for precision in part of the analysis. Thus, here we explain the derivation of our sensitivity and precision formulas in detail.

```
# BLASTN 2.2.25+
# Query: NC_003997.3-1994/1
# Database: /data/ncbi/blast/db/nt
# Fields: query id, subject id, % identity, alignment length, mismatches, gap opens, q. start, q. end, s. start, s. end, evalue, bit score, taxon
# 372 hits found
NC_003997.3-1994/1    gi|1043168508|gb|CP012728.1|    100.00  250    0    0    1    250    1704042 1703793 2e-123    452    Bacillus anthracis
NC_003997.3-1994/1    gi|1043163212|gb|CP012727.1|    100.00  250    0    0    1    250    1704042 1703793 2e-123    452    Bacillus anthracis
NC_003997.3-1994/1    gi|1043157917|gb|CP012726.1|    100.00  250    0    0    1    250    1704041 1703792 2e-123    452    Bacillus anthracis
NC_003997.3-1994/1    gi|1043152620|gb|CP012725.1|    100.00  250    0    0    1    250    1704043 1703794 2e-123    452    Bacillus anthracis
NC_003997.3-1994/1    gi|1043147323|gb|CP012724.1|    100.00  250    0    0    1    250    1704043 1703794 2e-123    452    Bacillus anthracis
NC_003997.3-1994/1    gi|589079678|gb|CP006742.1|     100.00  250    0    0    1    250    1690737 1690488 2e-123    452    Bacillus anthracis str. SVA11
NC_003997.3-1994/1    gi|570716017|gb|CP001970.1|     100.00  250    0    0    1    250    1703973 1703724 2e-123    452    Bacillus anthracis str. A16
NC_003997.3-1994/1    gi|570710095|gb|CP001974.1|     100.00  250    0    0    1    250    1703925 1703676 2e-123    452    Bacillus anthracis str. A16R
NC_003997.3-1994/1    gi|384383725|gb|CP002091.1|     100.00  250    0    0    1    250    1694749 1694500 2e-123    452    Bacillus anthracis str. H9401
NC_003997.3-1994/1    gi|229264291|gb|CP001598.1|     100.00  250    0    0    1    250    1703999 1703750 2e-123    452    Bacillus anthracis str. A0248
NC_003997.3-1994/1    gi|227002338|gb|CP001215.1|     100.00  250    0    0    1    250    2535117 2535366 2e-123    452    Bacillus anthracis str. CDC 684
NC_003997.3-1994/1    gi|218534755|gb|CP001283.1|     100.00  250    0    0    1    250    1762619 1762370 2e-123    452    Bacillus cereus AH820
...
```

- **bit score of the best hit to the target taxon (*B. anthracis*) = 452**
- **bit score of the best hit to a non-target taxon (*B. cereus*) = 452**
- **452 – 452 = 0; minimum required bit score difference determined by simulation = 8**
- **therefore, this read is not assigned to the target taxon (*B. anthracis*)**

**Figure 3** **Demonstration of the "bit score difference" filter.** In this first example, application of the bit score difference filter does not result in the assignment of the read to the target taxon.

## Calculation of sensitivity

To calculate sensitivity, one must determine the number of target taxon reads that were correctly assigned as a fraction of all the target taxon reads that were assigned. In this study, a true positive (TP) is a simulated read from the target taxon assigned correctly (either assigned directly to the target taxon or to a more specific taxon beneath the target), and a false negative (FN) is a simulated read from the target taxon assigned incorrectly (i.e., assigned to a taxon that is not part of the target taxon lineage). Note that the case of a non-specific but not incorrect read assignment (e.g., a *B. anthracis* read assigned to the *B. cereus group*) is neither considered a TP nor a FN; we term this an "inconclusive assignment" (IA). The count of TPs, FNs, and IAs can be easily determined by parsing the BLAST output associated with the target taxon. In all of our read simulation experiments, therefore, the calculation of sensitivity uses the formula (TP/(TP + FN + IA)).

## Calculation of precision

To calculate precision, one must determine the number of non-target taxon reads incorrectly assigned to the target taxon, each of which is considered a false positive (FP). Naively, determining the count of FPs would require simulating reads and evaluating BLAST results for every non-target taxon in the database, but we currently regard this as computationally prohibitive. Instead, we offer two alternatives. The first, which we call "near neighbor," computes FP using the genome in the database that is most globally

```
# BLASTN 2.2.25+
# Query: NC_003997.3-1992/1
# Database: /data/ncbi/blast/db/nt
# Fields: query id, subject id, % identity, alignment length, mismatches, gap opens, q. start, q. end, s. start, s. end, evalue, bit score, taxon
# 307 hits found
NC_003997.3-1992/1    gi|753450531|gb|CP009541.1|    100.00  250   0   0   1   250   322480 322231 2e-123  452   Bacillus anthracis str. Sterne
NC_003997.3-1992/1    gi|753270362|gb|CP009315.1|    100.00  250   0   0   1   250   4108350 4108599 2e-123 452   Bacillus anthracis str. Turkey32
NC_003997.3-1992/1    gi|673992625|gb|CP007666.1|    100.00  250   0   0   1   250   1176391 1176640 2e-123 452   Bacillus anthracis str. Vollum
NC_003997.3-1992/1    gi|589079678|gb|CP006742.1|    100.00  250   0   0   1   250   2289646 2289895 2e-123 452   Bacillus anthracis str. SVA11
NC_003997.3-1992/1    gi|570716017|gb|CP001970.1|    100.00  250   0   0   1   250   2303524 2303773 2e-123 452   Bacillus anthracis str. A16
NC_003997.3-1992/1    gi|570710095|gb|CP001974.1|    100.00  250   0   0   1   250   2303013 2303262 2e-123 452   Bacillus anthracis str. A16R
NC_003997.3-1992/1    gi|384383725|gb|CP002091.1|    100.00  250   0   0   1   250   2293805 2294054 2e-123 452   Bacillus anthracis str. H9401
NC_003997.3-1992/1    gi|229264291|gb|CP001598.1|    100.00  250   0   0   1   250   2303542 2303791 2e-123 452   Bacillus anthracis str. A0248
NC_003997.3-1992/1    gi|227002338|gb|CP001215.1|    100.00  250   0   0   1   250   1935455 1935206 2e-123 452   Bacillus anthracis str. CDC 684
NC_003997.3-1992/1    gi|49176966|gb|AE017225.1|     100.00  250   0   0   1   250   2303569 2303818 2e-123 452   Bacillus anthracis str. Sterne
NC_003997.3-1992/1    gi|30260185|gb|AE016879.1|     100.00  250   0   0   1   250   2303518 2303767 2e-123 452   Bacillus anthracis str. Ames
NC_003997.3-1992/1    gi|675832311|gb|CP008853.1|    99.60   250   1   0   1   250   2303095 2303344 9e-122  446   Bacillus anthracis
NC_003997.3-1992/1    gi|755995789|gb|CP009605.1|    99.20   250   2   0   1   250   1841890 1841641 1e-120  443   Bacillus cereus
...
```

- bit score of the best hit to the target taxon (*B. anthracis*) = 452

- bit score of the best hit to a non-target taxon (*B. cereus*) = 443

- 452 – 443 = 9; minimum required bit score difference determined by simulation = 8

- therefore, this read is assigned to the target taxon (*B. anthracis*)

**Figure 4 Demonstration of the "bit score difference" filter.** In this second example, application of the bit score difference filter results in the assignment of the read to the target taxon.

similar to the target taxon as a proxy for all non-target database taxa. The intuition behind this approach is that a misclassified read (presumably due to sequencing error) is most likely to originate from a database genome that is very similar to the target taxon. Thus, with the near neighbor approach, the calculation of precision uses the formula (TP/(TP + FP)). The potential weakness of this approach is that there could be a region of local similarity to the target taxon in a database genome that is not the near neighbor. Thus, we offer a second approach that does not rely on selecting other genomes from the database, which we call the "FNs" approach. This approach relies on the observation that if the sequencing error process is symmetric—i.e., the probability of an erroneous A to C substitution is the same as that of C to A, insertions are as probable as deletions, and so on—then the process that gives rise to FNs can be treated as equivalent to the process that gives rise to FPs. While it is known that in practice this assumption of symmetry is violated (*Schirmer et al., 2016*), it may nonetheless suffice to use FN as a proxy for FP in this context. Thus, with the FNs approach, it is only necessary to simulate reads from the target taxon, and the calculation of precision uses the formula (TP/(TP + FN)). Unfortunately, deciding which of the two heuristics is more effective would require comparison to a provably optimal procedure; in this study, we present results from simulated read experiments using both the near neighbor and FNs approaches, and report the patterns we observe.

## BLAST result parsing, final taxonomic assignment, and calculation of statistics

BLAST result parsing and final taxonomic assignment of each read was performed with a custom Perl script capable of querying the NCBI taxonomy database (*NCBI Resource Coordinators, 2016*). If a target taxon is supplied as an argument to the script, assignments to the target taxon lineage that are more specific than the target taxon are simply reassigned to the target taxon. BLAST hits that do not meet the criteria specified by the absolute filters (minimum alignment length, minimum alignment percent identity, or maximum *E*-value) are removed, as are hits to the "other sequences" clade (NCBI taxon ID 28384), which are presumed to be erroneous. To make the final taxonomic assignment for each read, the lowest common ancestor (LCA) algorithm (*Huson et al., 2007*) is applied to the remaining hits that have a difference in bit score from the best hit less than a specified amount. If multiple parameter values are supplied for one or more filters, the script parses the BLAST results once for each possible combination of parameter values and writes the results to separate "LCA files," thus enabling the user to efficiently perform parameter sweeps. The ultimate output from the script is one or more LCA files, each containing the final taxonomic assignment of each read for a particular combination of filter parameter values. Counts of TPs, FNs, IAs, and FPs (from which sensitivity and precision were calculated) were obtained using a separate Perl script that parses the LCA files produced by the BLAST result parser.

## Determination of optimal BLAST filter parameter values

When deciding how the absolute and relative BLAST filters should be parameterized, an optimality criterion is needed. In the execution of bioforensic casework, it is important that any assignments made are correct. Thus, we first chose filter parameter values that maximized precision (i.e., minimized incorrect read assignments). In the event that multiple combinations of parameter values yielded exactly the same maximum precision value, we chose from among these the combination that maximized sensitivity (i.e., maximized detection of the target taxon). In the event that multiple combinations of parameter values yielded exactly the same maximum precision *and* sensitivity values, we reported the strictest combination.

In this study, we present examples aimed at detecting a variety of pathogenic target taxa including *B. anthracis*, *Clostridium botulinum*, pathogenic *Escherichia coli*, and *Yersinia pestis*. Due to inherent variation in the degree of interrelatedness among genomes from different taxonomic clades, optimal filter parameter values need to be set differently for each target taxon. To determine optimal filter settings, we must know the true origin of our test sequences; thus, we simulate reads from each target taxon genome, BLAST them against `nt`, and evaluate classification performance under different combinations of filter parameter settings. In each read simulation experiment, 81 different combinations of filter parameter values were tested—i.e., all combinations of maximum *E*-value ($E$) = $\{10^0, 10^{-1}, 10^{-2}, 10^{-4}, 10^{-8}, 10^{-16}, 10^{-32}, 10^{-64}, 10^{-128}\}$ and bit score difference ($b$) = $\{0, 1, 2, 4, 8, 16, 32, 64, 128\}$. The parameter optimization workflow is shown in Fig. 5.
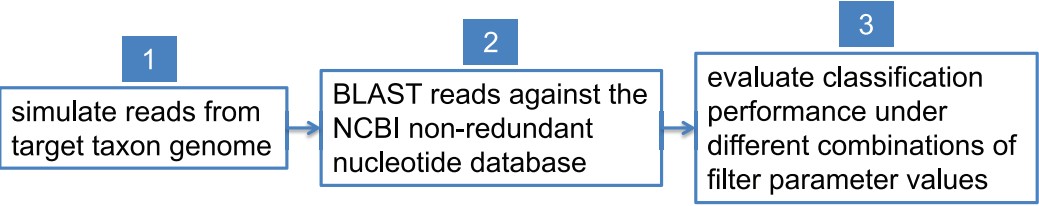

true positive (**TP**) = a simulated read from the underline{target} taxon assigned underline{correctly}

false negative (**FN**) = a simulated read from the underline{target} taxon assigned underline{incorrectly}

inconclusive assignment (**IA**) = a simulated read from the underline{target} taxon assigned underline{non-specifically}

sensitivity = (**TP** / (**TP** + **FN** + **IA**))

false positive (**FP**) = a simulated read from a underline{non-target} taxon underline{incorrectly} assigned to the underline{target} taxon

precision = (**TP** / (**TP** + **FP**))[1] underline{OR} (**TP** / (**TP** + **FN**))[2]

[1] Standard definition of precision, used with the near neighbour approach.

[2] Non-standard definition of precision where *FN* are used as a surrogate for *FP*, used with the false negatives approach.

**Figure 5** **Workflow for determining optimal BLAST filter parameter values.** Simulated reads from the target taxon genome are compared against the NCBI nt database using BLAST, and classification performance is evaluated under different combinations of parameter values used to filter BLAST results.

## Selection of near neighbor and alternate representative genomes

For each target taxon, we used the "Genome neighbor report" feature of the NCBI Genome database (*NCBI Resource Coordinators, 2016*) to select the most closely related complete genome of a different species or strain, as appropriate, to be used as the "near neighbor." For species-level target taxa, we used the Genome neighbor report to select the complete genome of the same species that was most distantly related to the original representative genome, which we call the "alternate representative genome" (Table 1).

## Clade-level exclusion

In the final read simulation experiment, clade-level exclusion (*Brady & Salzberg, 2009*) was performed to assess classification performance in the situation where the taxon for which one has sequence data is not represented in the reference database. In these tests, we simulated 250-bp reads from the taxon hypothetically missing from reference database, excluded this taxon from the reference database when performing BLAST searches, and then obtained optimal filter parameter values for classification of the target taxon, which in this case was the taxon immediately above the excluded taxon in taxonomic rank.

## RESULTS AND DISCUSSION

### Evaluation of a "first-pass" taxonomic classifier

To demonstrate typical use of a first-pass taxonomic classification program, we analyzed all simulated reads from the *B. cereus* JEM-2 genome (*Venkateswaran et al., 2017a*, *2017b*) with Kraken. The majority of the reads (79%) were assigned to the *B. cereus* group; of these, only 32% of the reads were assigned more specifically to *B. cereus*. Worryingly, however, a relatively small number of reads were assigned incorrectly to other *B. cereus*

**Table 1** Taxonomic data and metadata for target taxa and near neighbor species or strains.

| Taxon | Taxonomic rank | Type | NCBI Taxonomy ID | RefSeq assembly accession |
|---|---|---|---|---|
| *Bacillus cereus* group | Species group | Target | 86661 | N/A |
| *B. anthracis* | Species | Target | 1392 | N/A |
| *B. cereus* | Species | Near neighbor | 1396 | N/A |
| *B. anthracis* Ames Ancestor | Strain | Representative | 261594 | GCF_000008445 |
| *B. anthracis* Cvac02 | Strain | Representative (alternate) | N/A | GCF_000747335 |
| *B. cereus* JEM-2 | Strain | Representative | N/A | GCF_001941925 |
| *Clostridium* | Genus | | 1485 | N/A |
| *C. botulinum* | Species | Target | 1491 | N/A |
| *C. sporogenes* | Species | Near neighbor | 1509 | N/A |
| *C. botulinum* A str. ATCC 3502 | Strain | Representative | 413999 | GCF_000063585 |
| *C. botulinum* B1 str. Okra | Strain | Representative (alternate) | 498213 | GCF_000019305 |
| *C. sporogenes* NCIMB 10696 | Strain | Representative | N/A | GCF_000973705 |
| *Escherichia* | Genus | | 561 | N/A |
| *E. coli* | Species | | 562 | N/A |
| *E. coli* O157:H7 str. Sakai | Strain | Target | 386585 | GCF_000008865 |
| *E. coli* SRCC 1675 | Strain | Near neighbor | N/A | GCF_001612495 |
| *Yersinia pseudotuberculosis* complex | Species group | | 1649845 | N/A |
| *Y. pestis* | Species | Target | 632 | N/A |
| *Y. pseudotuberculosis* | Species | Near neighbor | 633 | N/A |
| *Y. pestis* CO92 | Strain | Representative | 214092 | GCF_000009065 |
| *Y. pestis* Angola | Strain | Representative (alternate) | 349746 | GCF_000018805 |
| *Y. pseudotuberculosis* PB1/+ | Strain | Representative | 502801 | GCF_000834475 |

group species, including *B. anthracis*, *B. cytotoxicus*, *B. mycoides*, *B. thuringiensis*, and *B. weihenstephanensis*. Had this benign strain of *B. cereus* (JEM-2) been the sole representative of the *B. cereus* group in a metagenomic sample, an analyst using Kraken might have erroneously declared that a variety of *B. cereus* group species were present in the sample, including pathogenic *B. anthracis*. As false-positive assignments are relatively commonplace with first-pass classification programs, we were motivated to develop a procedure to validate the assignments of interest made by such classifiers.

## Taxon selection

To demonstrate the BLAST-based validation procedure, we selected four target taxa, all of which are biological agents that could conceivably be of interest in a bioforensic setting. The first is *B. anthracis*, the bacterium that causes anthrax. The second is *C. botulinum*, a bacterium capable of producing the lethal botulinum neurotoxin. The third is a pathogenic strain of *E. coli*, *E. coli* O157:H7 str. Sakai, a bacterium that has been associated with major outbreaks of foodborne illness. The fourth and final target taxon is *Y. pestis*, the bacterium that causes bubonic plague. Thus, three out of the four target taxa represent particular species to be identified (*B. anthracis*, *C. botulinum*, and *Y. pestis*), whereas one target taxon represents a particular strain to be identified (*E. coli* O157:H7 str. Sakai). Species-level evaluations were performed using the representative strains indicated in Table 1.

**Table 2 Experiment 1: simulated 250-bp reads from four target taxa.**

| Target taxon | Taxonomic rank | Number of reads | Approach used to compute FP | Maximum E-value | Bit score difference | Validated reads | Sensitivity | Precision |
|---|---|---|---|---|---|---|---|---|
| B. anthracis | Species | 220,140 | Near neighbor | $10^{-64}$ | 8 | 19,491 | 0.088539 | 1.0 |
| | | | False negatives | $10^{-64}$ | 8 | 19,491 | 0.088539 | 0.999995[1] |
| | | | False negatives | $10^{-64}$ | 128 | 1,751 | 0.007954 | 1.0 |
| C. botulinum | Species | 156,120 | Near neighbor | $10^{-64}$ | 8 | 153,786 | 0.985050 | 1.0 |
| | | | False negatives | $10^{-64}$ | 8 | 153,786 | 0.985050 | 1.0 |
| E. coli O157:H7 str. Sakai | Strain | 223,760 | Near neighbor | $10^{-64}$ | 1 | 838 | 0.003745 | 1.0 |
| | | | False negatives | $10^{-64}$ | 8 | 184 | 0.000822 | 1.0 |
| Y. pestis | Species | 193,170 | Near neighbor | $10^{-64}$ | 8 | 20,398 | 0.105596 | 1.0 |
| | | | False negatives | $10^{-64}$ | 8 | 20,398 | 0.105596 | 1.0 |

**Notes:**
Optimal parameter values for filtering BLAST results were chosen to maximize precision (first) and sensitivity (second) using two different approaches to compute false positives.

[1] In the case of *B. anthracis*, we observe that sensitivity increased from ≈0.8% to ≈8.9% when allowing exactly one FN assignment (precision ≈ 99.9995%).

In two of the three evaluations, the genome chosen was a reference genome for the species (*C. botulinum* A str. ATCC 3502 and *Y. pestis* CO92). For the *B. anthracis* evaluation, the genome of the Ames Ancestor strain was used to ensure that the pXO plasmids were included, as presence of the pXO plasmids is normally required for *B. anthracis* to be fully virulent (*Okinaka et al., 1999a*, *1999b*; *Pannucci et al., 2002*). To evaluate the implications of representative genome choice, an alternate representative genome was selected for each species. Additional information about the target taxa and near neighbors is provided in Table 1.

## Simulated read experiments

A total of four simulated read experiments were performed to determine optimal BLAST filter parameter values for the identification of various target taxa. A comparison of sensitivity across experiments on a per-taxon basis is available in Figs. S1–S4, online.

### Experiment 1: 250-bp simulated reads

The first experiment simulated 250-bp reads from the target and near neighbor genomes; the results are shown in Table 2.

We observe that when requiring perfect precision, sensitivity was highest for identification of *C. botulinum* (≈99%), followed by much lower sensitivity for *B. anthracis* and *Y. pestis*. These results are understandable, as it is well established that the species that comprise *B. cereus sensu lato* have very similar genomic content (*Bazinet, 2017*), and that *Y. pestis* and *Y. pseudotuberculosis* are also very closely related (*Achtman et al., 1999*). Sensitivity was lowest for identification of *E. coli* O157:H7 str. Sakai (≈0.4% for near neighbor and ≈0.08% for FNs). Again, this result is consistent with the expectation that strain-level identification would be substantially more challenging than species-level identification, as the two *E. coli* strains in this case are ≈99.97% identical. Because the reads in this experiment were simulated from genomes that were present in the reference database, almost all read alignments had equally good scores, so the absolute *E*-value filter

**Table 3 Experiment 2: simulated 250-bp reads from three target taxa using alternate representative genomes.**

| Target taxon | Taxonomic rank | Number of reads | Approach used to compute FP | Maximum E-value | Bit score difference | Validated reads | Sensitivity | Precision |
|---|---|---|---|---|---|---|---|---|
| B. anthracis | Species | 209,080 | Near neighbor | $10^{-64}$ | 8 | 20,114 | 0.096202 | 1.0 |
| | | | False negatives | $10^{-64}$ | 8 | 20,114 | 0.096202 | 1.0 |
| C. botulinum | Species | 164,270 | Near neighbor | $10^{-64}$ | 8 | 162,069 | 0.986601 | 1.0 |
| | | | False negatives | $10^{-64}$ | 4 | 162,594 | 0.989797 | 1.0 |
| Y. pestis | Species | 187,470 | Near neighbor | $10^{-64}$ | 8 | 22,576 | 0.120425 | 1.0 |
| | | | False negatives | $10^{-64}$ | 8 | 22,576 | 0.120425 | 1.0 |

Note:
Optimal parameter values for filtering BLAST results were chosen to maximize precision (first) and sensitivity (second) using two different approaches to compute false positives.

had little or no effect until it was set so stringently that it eliminated all TP ($E = 10^{-128}$). In the case of *B. anthracis*, we observe that sensitivity increased from ≈0.8% to ≈8.9% when allowing exactly one FN assignment (precision ≈99.9995%; Table 2). This suggests that if one is willing to relax the perfect precision requirement very slightly, it may be possible to make significant gains in sensitivity. Finally, it is interesting to note that in most cases, $b = 8$ maximized sensitivity while achieving perfect precision. This likely represented a "sweet spot" (at least as compared to $b = 4$ or $b = 16$) for the level of taxonomic specificity represented by the selected target taxa.

### Experiment 2: 250-bp simulated reads, alternate representative genome

Choosing a particular genome to represent a strain, species, or higher-level taxon could in principle have implications for the filter parameter values recommended by the optimization procedure. While hopefully the taxonomy is structured such that members of a particular clade are more similar to each other than to members of other clades, taxonomies are well known to be imperfect in this regard. To test the implications of representative genome choice, we repeated the species-level evaluations from Experiment 1, except that we used an alternate representative genome for the target taxon, the database genome that was most distantly related to the original representative genome. The results are shown in Table 3.

In general, the optimal parameter values recommended by this experiment and the resulting values of sensitivity and precision were highly concordant with the results of Experiment 1 (Tables 2 and 3). The optimal parameter values recommended for classification of *B. anthracis* when using the Cvac02 strain were identical to those recommended when using the Ames Ancestor strain ($E = 10^{-64}$ and $b = 8$), with the exception that it was possible to achieve perfect precision using the FNs approach when $b = 8$. Likewise, when using *C. botulinum* B1 str. Okra, the FNs approach recommended $b = 4$ rather than $b = 8$. These results suggest that the filter parameter values recommended by the FNs approach are potentially more dependent on representative genome choice than those recommended by the near neighbor approach. The calculation of FP in the FNs approach is based solely on the classification of reads simulated from the chosen representative genome, whereas in the near neighbor approach, FP can result from the assignment of a near neighbor read to any genome associated with the target

**Table 4 Experiment 3: simulated 150-bp reads from four target taxa.**

| Target taxon | Taxonomic rank | Number of reads | Approach used to compute FP | Maximum E-value | Bit score difference | Validated reads | Sensitivity | Precision |
|---|---|---|---|---|---|---|---|---|
| B. anthracis | Species | 366,920 | Near neighbor | $10^{-32}$ | 8 | 16,904 | 0.046070 | 1.0 |
| | | | False negatives | $10^{-32}$ | 8 | 16,904 | 0.046070 | 1.0 |
| C. botulinum | Species | 260,200 | Near neighbor | $10^{-32}$ | 8 | 250,915 | 0.964316 | 1.0 |
| | | | False negatives | $10^{-32}$ | 8 | 250,915 | 0.964316 | 1.0 |
| E. coli O157:H7 str. Sakai | Strain | 372,960 | Near neighbor | $10^{-64}$ | 4 | 709 | 0.001901 | 1.0 |
| | | | False negatives | $10^{-64}$ | 8 | 180 | 0.000483 | 1.0 |
| Y. pestis | Species | 321,970 | Near neighbor | $10^{-32}$ | 8 | 19,965 | 0.062009 | 1.0 |
| | | | False negatives | $10^{-32}$ | 8 | 19,965 | 0.062009 | 1.0 |

**Note:**
Optimal parameter values for filtering BLAST results were chosen to maximize precision (first) and sensitivity (second) using two different approaches to compute false positives.

taxon (any strain of *B. anthracis*, for example). Thus, it might behoove a user of our method to sample the diversity in their clade of interest by running the optimization procedure for multiple representatives and using the globally most conservative recommended parameter values for classification (if maximizing precision is the goal). Alternatively, one might devise a method for more exhaustive sampling of the diversity that might exist among target taxon genomes.

### Experiment 3: 150-bp simulated reads

Experiment 3 was identical to Experiment 1, except that a simulated read length of 150 bp was used, thus making the classification task more difficult. The results are shown in Table 4.

With optimal filter parameter values, we observe that sensitivity in detecting each target taxon decreased relative to the 250-bp experiment—e.g., in the case of *C. botulinum*, sensitivity decreased from ≈99% to ≈96% (Tables 2 and 4). Also, optimal values for the *E*-value and bit score difference filters varied somewhat relative to the 250-bp experiment, although it was always the case that $E \leq 10^{-64}$ and $b \leq 8$.

### Experiment 4: clade-level exclusion, 250-bp simulated reads

In a final simulated read experiment, clade-level exclusion of either species (*B. anthracis*) or strains (*C. botulinum* A str. ATCC 3502 and *Y. pestis* CO92) was performed to assess classification performance when the taxon for which one has sequence data is not represented in the reference database, a situation commonly encountered in practice. Only the FNs method of computing FP was used; the results are shown in Table 5.

In this experiment, we observe that it was not always possible to achieve perfect precision—maximum precision for identification of the *B. cereus* group when excluding *B. anthracis* was ≈93.9%, and maximum precision for identification of *C. botulinum* when excluding *C. botulinum* A str. ATCC 3502 was ≈99.9%. We note that sensitivity for identification of *C. botulinum* decreased from ≈98.5% in Experiment 1 (Table 2) to ≈90.5% in the clade-level exclusion experiment (Table 5). By contrast, sensitivity for identification of *Y. pestis* hardly decreased at all (≈10.6% vs. ≈10.5%).

**Table 5 Experiment 4: simulated 250-bp reads from three taxa that were summarily excluded from the reference database.**

| Target taxon | Taxonomic rank | Excluded taxon | Number of reads | Maximum $E$-value | Bit score difference | Validated reads | Sensitivity | Precision |
|---|---|---|---|---|---|---|---|---|
| *B. cereus* group | Species group | *B. anthracis* | 220,140 | $10^{-64}$ | 64 | 36,003 | 0.163546 | 0.939339 |
| *C. botulinum* | Species | *C. botulinum* A str. ATCC 3502 | 156,120 | $10^{-64}$ | 32 | 141,277 | 0.904926 | 0.999385 |
| *Y. pestis* | Species | *Y. pestis* CO92 | 193,170 | $10^{-64}$ | 8 | 20,272 | 0.104944 | 1.0 |

**Note:**
Optimal parameter values for filtering BLAST results were chosen to maximize precision (first) and sensitivity (second) using the "false negatives" approach to compute false positives.

## Calculating precision: "near neighbor" vs. "false negatives"

Our results show that it was possible to achieve perfect precision in 3/4 simulated read experiments when using either the near neighbor or the FNs approach (the exception being the clade-level exclusion experiment). In 7/11 cases, the filter parameter values recommended by the two approaches were identical; in the cases where they differed, the FNs approach uniformly recommended more stringent filter parameter values than the near neighbor approach, resulting in reduced sensitivity (Tables 2–4). As mentioned previously, deciding which of the two approaches to calculating precision is superior would require a comparison to a provably optimal approach, which we currently deem computationally intractable. Each heuristic makes assumptions that may not always hold: the near neighbor approach assumes that a single genome that is closely related to the target taxon is sufficient to serve as a proxy for all other non-target taxa in the database, and the FNs approach assumes that the sequencing error process is symmetric. When seeking to avoid an erroneous claim that a particular biological agent is present in a sample, one may wish to use the more conservative set of parameter values recommended by the two approaches.

## Practical application of the BLAST-based validation procedure

To demonstrate the practical application of the BLAST-based validation procedure, we downloaded a subset of metagenomic data collected from the New York City subway system (NCBI SRA ID SRR1748708), which the original study indicated might contain some reads from *B. anthracis* (Afshinnekoo et al., 2015). Indeed, analysis of this data with Kraken, our first-pass classifier, assigned 676 reads to *B. anthracis* (≈0.04% of reads). However, BLAST-validation of these 676 reads using the most conservative parameters recommended by our study ($E = 10^{-64}$ and $b = 128$; Table 2) resulted in zero reads assigned to *B. anthracis*. Even after significantly relaxing the minimum required bit score difference (setting $b = 8$), which was shown in Experiment 1 to significantly increase sensitivity (Table 2), still zero reads were assigned to *B. anthracis*. Thus, we would conclude that the 676 reads that Kraken assigned to *B. anthracis* were in fact false-positive assignments, which agrees with other follow-up studies that have been performed on the New York City subway data (Gonzalez et al., 2016).

## CONCLUSION

We have shown how BLAST, a very widely used tool for sequence similarity searches, can be used to perform taxonomic assignment with maximal precision by using BLAST result filters fine-tuned via read simulation experiments in conjunction with an LCA algorithm. We demonstrated the parameter optimization process for four different pathogenic organisms, and showed that optimal parameter values and resulting values of sensitivity and precision varied significantly depending on the selected taxon, taxonomic rank, read length, and representation of the sequenced taxon in the reference database. Furthermore, the addition or removal of a single sequence from the reference database could change the recommended optimal parameter values, so the optimization process should be re-run every time the database is updated.

Once optimal BLAST filter parameter values for a particular taxon have been determined, they can be subsequently used to perform validation of sequence assignments to that taxon. Given the massive size of many metagenomic data sets, however, we envision most users employing a "two-step" approach that involves first producing candidate target taxon sequence assignments using a relatively fast classification program—one that is not necessarily optimized for precision—and then confirming the veracity of those sequence assignments using the BLAST-based validation procedure.

One would be hard-pressed to define a "typical" metagenomic experiment, and the probability that a particular genome that is physically present in a metagenomic sample at some abundance is ultimately represented in the sequencing library and sequenced to a particular degree of coverage is a function of many factors that are outside the scope of this study. The methods we present here are concerned with read-by-read taxonomic assignment (each read interrogated independently of all other reads), and the selection of optimal BLAST result filters for this assignment process—in our case, we define "optimal" to mean correct assignment of the greatest possible number of reads without any incorrect assignments. In a real-world detection scenario, an additional question will often be asked: how many reads should be assigned to a particular target taxon before one deems it "present" in the sample? In principle, if one assumes that the reads in question originate from a genome that is present in the reference database, and that there was no error associated with the read simulation process or choice of optimal filter parameter values, then the answer is simply "one read." In practice, however, if only one read out of millions or billions is assigned to a particular taxon, it is only natural that one may hesitate to claim that a potential pathogen or other biological agent is present in a sample on the basis of such scanty evidence. Unfortunately, meaningful additional guidance on this point would require a comprehensive accounting of all possible sources of error associated with the analysis of a metagenomic sample.

Potential users of the software will find scripts for parsing BLAST results, performing parameter sweeps, and assigning final taxon labels to sequences at https://github.com/bioforensics/blast-validate.

## ACKNOWLEDGEMENTS

We thank Todd Treangen and Nicholas Bergman for helpful discussions about the project, and Travis Wyman, Timothy Stockwell, and Lisa Rowe for providing feedback on drafts of the manuscript.

The views and conclusions contained in this document are those of the authors and should not be interpreted as necessarily representing the official policies, either expressed or implied, of the DHS or S&T. In no event shall DHS, NBACC, S&T or Battelle National Biodefense Institute have any responsibility or liability for any use, misuse, inability to use, or reliance upon the information contained herein. DHS does not endorse any products or commercial services mentioned in this publication.

### Funding

This work was funded under Contract No. HSHQDC-15-C-00064 awarded by the Department of Homeland Security (DHS) Science and Technology Directorate (S&T) for the operation and management of the National Biodefense Analysis and Countermeasures Center (NBACC), a Federally Funded Research and Development Center. The funders had no role in study design, data collection and analysis, decision to publish, or preparation of the manuscript.

### Grant Disclosures

The following grant information was disclosed by the authors:
Department of Homeland Security (DHS) Science and Technology Directorate (S&T), National Biodefense Analysis and Countermeasures Center (NBACC):
HSHQDC-15-C-00064.

### Competing Interests

The authors declare that they have no competing interests.

### Author Contributions

- Adam L. Bazinet conceived and designed the experiments, performed the experiments, analyzed the data, contributed reagents/materials/analysis tools, prepared figures and/or tables, authored or reviewed drafts of the paper, approved the final draft.
- Brian D. Ondov conceived and designed the experiments, performed the experiments, analyzed the data, contributed reagents/materials/analysis tools, authored or reviewed drafts of the paper, approved the final draft.
- Daniel D. Sommer conceived and designed the experiments, performed the experiments, analyzed the data, contributed reagents/materials/analysis tools, authored or reviewed drafts of the paper, approved the final draft.
- Shashikala Ratnayake conceived and designed the experiments, performed the experiments, analyzed the data, authored or reviewed drafts of the paper, approved the final draft.

## Data Availability

All genome data used in this study is available from the NCBI RefSeq database (*O'Leary et al., 2015*) (assembly accessions provided in Table 1). The software implementing the methods described in this study, as well as simulated reads, analysis results, and other files germane to this study are available online at: https://github.com/bioforensics/blast-validate.

## Supplemental Information

Supplemental information for this article can be found online at http://dx.doi.org/10.7717/peerj.4892#supplemental-information.

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
