# Peer review of "BLAST-based validation of metagenomic sequence assignments"

_PeerJ, doi:10.7717/peerj.4892_

## Round 0.1 · original submission · Major Revisions

I have now received two thorough reviews of your paper. Both reviewers found the study well done and the research of broad interest. However, both also have a number of concerns and have made a number of suggestions to help you improve your paper. They have provided very good reviews, so I will not repeat their issues here. I hope you find them helpful in revising your paper.

Reviewer 1 ·

Basic reporting

Bazinet et al. developed a method for performing targeted strain level classification of sequencing reads from a metagenomic sample. The proposed method uses a less precise taxonomic classification algorithm as a first pass, e.g. Kraken, to identify candidate reads for validation. The authors demonstrate their method using simulated sequence data from four organisms relevant to the biothreat community. Primary issues with the manuscript are:
1. Experimental methods are hard to understand as present,
2. TP, FN, sensitivity, and precision definitions are misleading, and
3. lack of application.

These issues are covered in the experimental design and validity of the findings sections.

Overall the manuscript is well written, though the article would benefit from; restructuring, additional references, consolidated figures and tables along with more descriptive legends, and a number of sentences throughout the manuscript could be stated more succinctly.

The article could be restructured as text describing the methods and experimental design are distributed throughout the manuscript. Additionally, the section on related methods is out of place. Method description is scattered throughout the manuscript. Moving all methods to the methods section as well as having separate sections describing the blast-validate method (parameter sweep and BLAST result filtering) and method evaluation would help clarify what was done. Additionally, the clade level exclusion method is only briefly described in the methods section without sufficient context for the reader to know how and why the method is used in the study. Providing scripts used to perform the method evaluation, genome sequences, and seed numbers used to simulate the sequencing reads would facilitate study reproducibility.
I recommend adding references for other taxonomic classification methods developed for strain and species level classification. Additionally, while the proposed method is a targeted approach the introduction would benefit from the inclusion of large taxonomic classification benchmarking studies, e.g. McIntyre et al. 2017, and Sczyrba et al 2017. Including these references would help establish the study motivation, potentially replacing the ‘Evaluation of a “first-pass” taxonomic classifier’ section.

The bulk of the study results are presented in four tables. Due to the overlap of information presented in the four tables, the tables could be combined. Combining the tables would make it easier for the reader to compare results across evaluation experiments. For most evaluations, the near-neighbor and false-negative methods for calculating precision are the same. Only providing results for evaluations where the values differ would make the tables more concise. Combining tables can help highlight show how different genomes (representative and alternative) impacts parameter threshold values. More descriptive figure and table legends would greatly help the reader interpret and digest the material. See experimental design and the general comments section for additional comments on the figures.

McIntyre, Alexa BR, et al. "Comprehensive benchmarking and ensemble approaches for metagenomic classifiers." Genome biology 18.1 (2017): 182.

Sczyrba, Alexander, et al. "Critical Assessment of Metagenome Interpretation—a benchmark of metagenomics software." Nature methods 14.11 (2017): 1063.

Experimental design

The manuscript represents original research that falls within the scope of the journal. Though the research question is not particularly well defined. The authors present a method for BLAST parameter optimization, with no application other than data used to define filter parameters. The study would benefit from processing additional datasets using the proposed optimized BLAST method to validate “first-pass” classifier results. This could be achieved using datasets where the target taxa is present, or simulating different target taxa levels by spiking in reads simulated from the target taxa into a dataset that does not contain the target taxa.

Method description is dispersed throughout the introduction, methods, and results sections making it hard piece together the experimental design during the first read through. Recommend having two methods sections, one describing the validation method, and a second describing method evaluation using simulated sequencing data. Additionally, a diagram for each section would provide additional clarity in differentiating the proposed blast-validate method the approach used to evaluate the method.

Two additional minor points, the clade-level exclusion method needs to be clearly described in the method section. The proposed evaluation figure would help. Additionally, the RefSeq release version should be stated in the methods along with the accession date.

Validity of the findings

Defining parameters to validate taxonomic assignments for specific taxa is a valuable tool relevant to the bioforensic as well as other application areas, e.g. pathogen detection. However, the sensitivity and specificity definitions are not standard and not well described. For example, the sensitivity definition used in the manuscript, TP/read count, is not the standard definition, and the alternative precision definition is the standard sensitivity definition. This leads the reader to question why optimizing for standard definitions of precision and sensitivity yields similar parameter values. Looking at the bast-validate source code, TP is defined as reads classified to the target taxa, and FN is any read not classified as the target taxa, the target taxa lineage, or no database hits. The four truth table values, TP, TN, FP, FN should add up to the total number of observations, here read counts. When the representative genome or alternative representative genomes are analyzed there are only TP or FN. Where TP are reads assigned to the target taxa and FN is defined as reads not assigned to the target taxa. These definitions for TP and FN is consistent with your definition and the standard definition of sensitivity, though inconsistent with both of your precision definitions. Conversely, all reads simulated from the near-neighbor genome can be classified as TN or FP. Using a combined dataset of simulated reads from the representative genome and near-neighbor you could obtain a complete set of truth table values for your parameter sweep evaluation. To address the issue of the near-neighbor not being representative of all potential false positives BLAST results for the representative genome could be used to identify a set of non-target taxa genomes to include as potential FPs and TNs in the dataset used for the parameter sweep.

The simulation results used to define the parameter values for filtering BLAST results to obtain perfect precision are then used to determine method sensitivity. There is a risk of overfitting the data when using the same set of simulated reads for performing the parameter sweep and sensitivity analysis. Expanding on the use of alternative reference genomes by performing a more formal cross-validation to assess parameter robustness as well as an example application as mentioned in the experimental design section would significantly strengthen the study.

Additional comments

- Of the taxonomic systems mentioned in the introduction and figure 1 two are for 16S database references and only one of which is relevant to the database used in the study. I recommend removing reference to different taxonomic systems in the introduction and figure 1 as there is minimal relevance to the study as presented.

- The authors include a number of quoted phrases, make sure new terminology is clearly defined. A section with new terms and definitions might be helpful.

- Provide a more detailed definition of BLAST bit score and E-value as understanding these values is key to understanding the BLAST-validation method.

- State pXO relevance and reference line 207

- The supplemental figures and tables present the same set of results. I recommend using the table to present the optimized parameter values and figures to show the “sensitivity” results. The figures would benefit from a more descriptive figure legend.

·

Basic reporting

I think the paper was well justified and clear. I strong believe that there is a need for this tool. In our results, we often find false results/assignments and often we need to remove them. I think the figures and table were high quality and useful.

One minor concern: the authors use experimental and simulations in the same lines to discuss their simulations. This is confusing, as I would associate "experimental" with real data, and not simulations.

Experimental design

The work the authors conducted was sound and clear. Their simulated read datasets clearly validate that their methods work the way they designed them. However, I think it is not sufficient to only use simulated reads.

I know their comment in the paper is that "there is no 'typical' metagenomic experiment." However, I think its important to show a few read-data examples. The simulated datasets show that for high-quality reads the approach works well. However, it is also crucial to show that the method can also clarify the results of real examples as well--even if you cannot definitively show that the answers are correct.

In addition, although the authors clearly show their approach significantly improves the Kraken result, it leaves me curious as to how well it improves other methods. (I of course would like to see how well our PathoScope approach fairs on the first-pass, and then how well much improvement the authors). I think they mention MEGAN, PhyloPhythia, Centrifuge in the paper. I would like to see them compare with PathoScope and MetaPhlAn as well.

There is no discussion of run times. I can imagine that the computation would not be prohibitive if the taxa of interest does not dominate the sample. When is this approach reasonable and when would it take too long? For example, a human fecal sample might be dominated by Bacteriodes, so using this approach to validate/refine these taxa would not be reasonable. Also, what is the tradeoff using slower but more accurate first-pass approaches (i.e. PathoScope) versus faster less accurate methods (i.e. Kraken) when combined with the proposed approach? (i.e. is PathoScope+BLAST roughly the same as Kraken+BLAST, but may be faster?)

I took a look at the author's software but I did not run it on a dataset. I think it would be helpful to provide more documentation and a concrete software example/tutorial.

Validity of the findings

I think the results and conclusions the authors make are well justified and sound. I would have just liked to see the authors address the comments in the previous section.

Additional comments

No major additional comments. I enjoyed reading this paper, and I believe it has the potential to make an impact on the field.

External reviews were received for this submission. These reviews were used by the Editor when they made their decision, and can be downloaded below.

---

## Round 0.2 · Minor Revisions

I have sent your paper back to one of the previous reviewers and there are still just a couple of minor issues to deal with. Otherwise, we agree that your paper is a great study and you've done an excellent job taking care of the points raised in the first round of review. This should be a quick turn around for you. Good luck with it.

Reviewer 1 ·

Basic reporting

No additional comment.

Experimental design

The NYC subway analysis should be added to the methods section.

Validity of the findings

No additional comment.

Additional comments

Dear Authors,
I commend you on thoroughly addressing the issues raised by myself and the other reviewer.

A few suggestions still remain.
1. For the related work section in the Introduction, a sentence stating that you will describe some related projects before presenting your new method will make the section fit the rest of the manuscript.
2. For the sensitivity and precision definitions, please state early on in the section that non-traditional formulas are used to calculate the metrics but the metric behavior is consistent with standard metrics. You may want to also emphasize that the standard metric equations are not easily applied to metagenomic taxonomic classification analysis and this is an issue that needs to be addressed by the research community.
3. Figure 1 - Greengenes, OTT, SILVA, and RDP are not taxonomies, but rather reference databases with sequences and taxonomic information. Please only include taxonomic systems such as NCBI (https://www.ncbi.nlm.nih.gov/taxonomy) and Bergy’s (https://www.bergeys.org/outlines.html).


Best Wishes

External reviews were received for this submission. These reviews were used by the Editor when they made their decision, and can be downloaded below.

---

## Round 0.3 · accepted · Accept

Thank you for your careful revision. I agree with your response to the last review. I especially appreciate your approach to the 'taxonomy' issue as clearly the reviewer was confused. I think your added discussion will help readers understand that these other systems are, indeed, taxonomies. I especially appreciate this since I helped work on the OTT and I KNOW it is a robust and unique taxonomy.

# External reviews were received for this submission. These reviews were used by the Editor when they made their decision, and can be downloaded below.